# Abductive Logical Rule Induction by Bridging Inductive Logic Programming and Multimodal Large Language Models

## Abstract

We propose ILP-CoT, a method that bridges Inductive Logic Programming (ILP) and Multimodal Large Language Models (MLLMs) for abductive logical rule induction. The task involves both discovering logical facts and inducing logical rules from a small number of unstructured textual or visual inputs, which still remain challenging when solely relying on ILP, due to the requirement of specified background knowledge and high computational cost, or MLLMs, due to the appearance of perceptual hallucinations. Based on the key observation that MLLMs could propose structure-correct rules even under hallucinations, our approach automatically builds ILP tasks with pruned search spaces based on the rule structure proposals from MLLMs, and utilizes ILP system to output rules built upon rectified logical facts and formal inductive reasoning. Its effectiveness is verified through challenging logical induction benchmarks, as well as a potential application of our approach, namely text-to-image customized generation with rule induction. Our code and data are released at
`https://anonymous.4open.science/r/ILP-CoT-Ano-83DC/`.

## 1 Introduction

Although remarkable progress has been made in improving the deductive reasoning abilities of AI systems (OpenAI, 2024b; Guo et al., 2025), inductive reasoning from raw data, which challenges more on perceiving and understanding complex raw inputs than long-chain deductive reasoning, remains a significant challenge. To pursue this direction, we study the task of abductive logical rule induction, in which the target is to utilize a small number of unstructured textual or visual instances to automatically identify and ground symbolic concepts and then inducing possible logical rules indicated by the instances. This reasoning task involves the dual challenges of both input perception and logical induction. On the one hand, the model needs to extract abstract and transferable symbolic concepts from input instances; on the other hand, utilizing limited instances, it must accurately infer the underlying logical relationships or rules.

Traditionally, abductive logical rule induction can be solved by a two-step pipeline. In the first step, a preprocessing process is performed for visual perception of the symbolic concepts. Afterwards, an external Inductive Logic Programming (ILP) module (Muggleton & De Raedt, 1994; Cropper & Dumančić, 2022) is introduced for logical rule induction. ILP systems are formal logical reasoning systems with strong advantages in terms of interpretability and verifiability. By inductively learning from a finite set of facts and background knowledge, ILP is able to produce logically transparent and auditable rules. From a theoretical perspective, the rules output by ILP can be formally verified, a feature that is particularly important in high-risk scenarios or applications with stringent correctness requirements. However, ILP also faces fundamental challenges, such as relying on structured input data and potential inefficiency in large-scale data settings. In response to these challenges, recent work has begun to explore neurosymbolic methods that integrate deep learning with symbolic logical reasoning (Evans & Grefenstette, 2018; Manhaeve et al., 2018; Dai & Muggleton, 2020; Cunnington et al., 2023; Shindo et al., 2023; 2024). These approaches attempt to use neural networks for perception and representation learning, then employ ILP or other logic-based modules for rule induction and inference. Even though these approaches significantly enlarge the applicability in

real applications, utilizing ILP usually requires to design logical background knowledge by human experts, which is a fundamental obstacle in handling challenging inductive reasoning problems.

With the rise of Multimodal Large Language Models (MLLMs) (OpenAI, 2024a; Liu et al., 2023a; Bai et al., 2023b; Wang et al., 2023b), researchers have begun to explore the application of these powerful models to textual and visual understanding and generation tasks. Due to training on massive datasets, MLLMs already exhibit astonishing performance in handling multimodal inputs, extracting symbolic representations, and mining rich semantic information, making them promising candidates for addressing abductive visual rule induction. However, MLLMs still face multiple bottlenecks in perception and reasoning (Zhang et al., 2023), including hallucination phenomena, highly opaque reasoning processes, and a lack of verifiable logical chains. We discover that these bottlenecks still limit the ability of MLLMs to directly solve abductive logical rule induction, even when guided by Chain-of-Thought (CoT) reasoning (Wei et al., 2022).

As a result, it is difficult to rely solely on traditional ILP approaches or MLLMs to achieve a balanced solution that is robust and interpretable in abductive logical rule abduction. In this work, we propose a hybrid method, ILP-CoT, to bring the best of both worlds. Our approach integrates the ILP system into the CoT reasoning process of MLLMs in a "plug-and-play" manner, forming a fully interpretable reasoning pipeline from start to finish without additional training. Specifically, we leverage the strong cross-modal perception and symbol extraction capabilities of MLLMs to automatically generate initial logical facts, i.e. logical predicates and background knowledge from the input instances, where perceptual hallucinations could exist. Afterwards, based on the key observation that MLLMs could propose structure-correct rules even under hallucinations, our approach introduces a deterministic conversion process to automatically transform the rule structure proposals from MLLMs into ILP meta-rules, realizing the key technical step of building ILP tasks with pruned search spaces. Finally, we dynamically invoke an ILP system to perform formal rule induction, yielding explainable and verifiable rules with rectified logical facts. This division of labor separates the complex symbolic grounding process and reduces the size of the rule hypothesis space, letting ILP focus exclusively on the more compact and structured symbolic data. This approach not only reduces the risk of hallucination during MLLM-driven inference, but also relies on formal verification from ILP to ensure the accuracy and consistency of the rules.

We introduce challenging CLEVR-Hans (Shindo et al., 2024) and ARC (Chollet, 2019; Xu et al., 2023) logical induction benchmarks to systematically verify the efficacy of our approach. Furthermore, we propose a potential application, namely text-to-image customized generation with rule induction. In this task, a small number of images provided by the user are given, including multiple subjects that the user cares about. Furthermore, the images are labeled by the user as "liked" or "disliked", followed by the preferences of the user for some latent regularities among the subjects, which can be represented by logical rules. We show that our approach enables to induce the latent logical rules from the training images, which can be utilized by downstream pre-trained text-to-image generation models to further generate images following user preferences.

## 2 RELATED WORK

Avoiding hallucination is a fundamental challenge for Large Language Models (LLMs) (Dasgupta et al., 2022; Saparov & He, 2022). A widely adopted strategy is Retrieval-Augmented Generation (RAG) (Lewis et al., 2020), which utilizes retrieval in external knowledge bases to avoid generating ungrounded contents. Although RAG is effective in reducing factual errors, it is invalid for rectifying fallacious reasoning processes. Furthermore, the requirement of accessing an external knowledge base is somehow limited for solving general reasoning tasks. Recently, there has been a growing trend in research to combine formal methods in LLMs. On the one hand, there have been attempts to use formal programming code (Gao et al., 2023; Li et al., 2023; Chae et al., 2024; Ling et al., 2024) or logical rules (Xu et al., 2024) as intermediate content to generate during CoT reasoning. These studies justify that formalizing the reasoning states can improve the accuracy of the reasoning chain without using external tools. However, it is still challenging to conduct fully reliable reasoning based on this mechanism. On the other hand, the idea of integrating external formal reasoning systems with LLMs has been explored in various reasoning tasks. Some research proposed to transform natural languages into code and further execute them using external symbolic solvers (Wu et al., 2022; He-Yueya et al., 2023; Lyu et al., 2023; Pan et al., 2023a; Ye et al., 2024; Jiang et al., 2024). A major issue is

that formalizing natural language into executable code is itself a difficult task, which is also a key challenge that we try to tackle in our work. In complex reasoning tasks with long reasoning chains, such as solving mathematical challenges, formal reasoning systems are treated as the sledgehammer to integrate with LLMs (Trinh et al., 2024). Unlike existing approaches that focus on deductive reasoning tasks (Pan et al., 2023b; Olausson et al., 2023), our work focuses on inductive reasoning. The significant difference lies in that inductive reasoning usually does not challenge the ability to do long-step reasoning but rather the ability to perceive and understand the input. This makes inductive reasoning more challenging for MLLMs due to the difficulty in perceiving complex multimodal inputs. We also note that this makes our contribution in parallel with multimodal deductive reasoning methods (Wang et al., 2022; Madaan et al., 2023; Gao et al., 2024; Mondal et al., 2024). The closest research to ours are Wang et al. (2023a); Qiu et al. (2023) to solve pure textual inductive reasoning. Their approach also uses LLMs to propose inductive hypothesis in Python and conduct program execution for correctness verification. In comparison, our approach utilizes a different methodology of bridging ILP reasoning and MLLMs to address logical induction tasks and alleviate hallucinations. Research on breaking the perceptual limitations of MLLMs has also received great attention. Existing approaches utilize scene graph knowledge (Mitra et al., 2023) or visual prompts (Wu et al., 2024), while formal methods have not received significant attention in MLLMs. The closest idea comes from visual question answering, in which textual LLMs are integrated with visual perception models to perform visual reasoning tasks (Hsu et al., 2024; Kamali et al., 2024). Purely textual LLMs rely on external visual processing models to perceive the input, while the target of our work is to conduct multimodal abductive induction based on the internal perceptual ability of MLLMs without using external tools to take the responsibility of perception.

## 3 ILP-CoT

### 3.1 PRELIMINARIES

In an abductive logical rule induction task, a small number of textual or visual instances are provided. Each instance is unstructured without any symbol-related annotations, while is labeled as *positive* or *negative* based on whether it is consistent with a set of latent logical rules, which describe regularities among multiple pre-defined subjects existing in all instances [1]. The targets are twofold: 1) transforming the unstructured instances into structured ones to discover the logical facts, i.e. symbolic concepts about the subjects involved in the latent logical rules, and their corresponding grounding values; 2) inducing the latent logical rules based on the discovered logical facts.

We introduce the ILP-CoT method bridging ILP and MLLMs to effectively solve the abductive logical rule induction tasks. Fig. 1 illustrates the workflow of ILP-CoT, which integrates the ILP system into the CoT reasoning process of MLLMs in a "plug-and-play" manner. For better understanding the technical design choice, we briefly introduce the reasoning mechanism of ILP systems following (Cropper & Dumančić, 2022), and refer the detailed introduction of ILP to this literature.

ILP seeks to identify a set of logical rules $H$ that can explain (or more formally, entail) all positive examples $E^+$ while excluding the optional negative examples $E^-$, based on background knowledge $B$ [2]. The positive and negative examples are sets of logical clauses. Each clause, representing one data instance, is of the form $p(x_1, x_2, \ldots, x_m)$, where each $x_i$ is a term representing a subject in the data, and $p$ is the predicate representing specific logical facts among all $x_i$. The background knowledge $B$ contains relations and information indirectly associated with the examples, which are also sets of logical clauses. To conduct rule induction, ILP systems follow the basic mechanism common in formal methods: searching in the hypothesis space $\mathcal{H}$ of all possible logical rules to identify $H$ satisfying the above target. Notably, the background knowledge $B$ can include clauses representing essential restrictions on the hypothesis space, in special rule structure constraints, to serve as the inductive bias of the hypothesis space. As in general machine learning problems, properly choosing the inductive bias would significantly prune the hypothesis space and improve the efficiency of induction. The basic idea in ILP-CoT is to let MLLMs play the rule of proposing initial (probably hallucinated) logical facts from the unstructured raw input images and more importantly, the proper rule structure constraints to build efficiently solvable ILP tasks, and further let the ILP system generate

---

[1]The negative instances are optional to exist in the task.

[2]We consider the learning from entailment (LFE) setting of ILP.

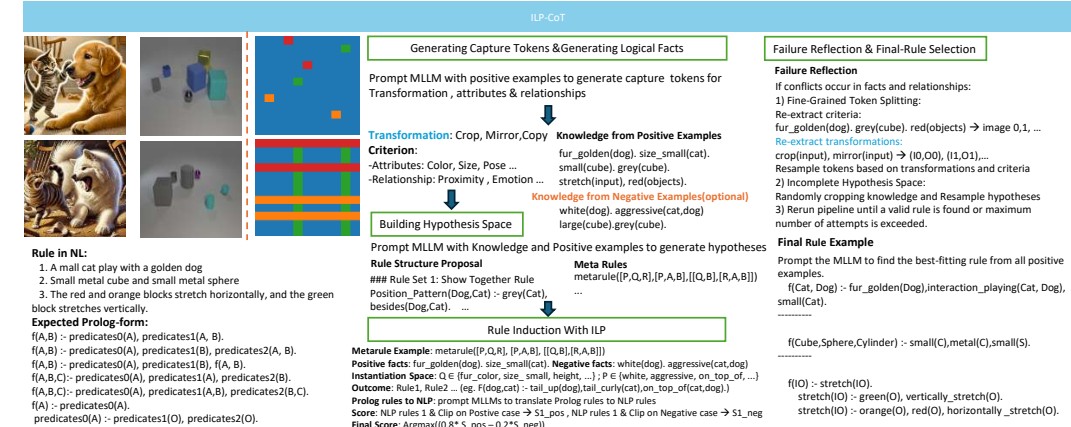

Figure 1: The ILP-CoT reasoning workflow.

the correct rules based on rectified logical facts with formal inductive reasoning. In the following, we dive into the details of each step of the reasoning process.

## 3.2 GENERATING INITIAL LOGICAL FACTS

We propose a unified formal procedure to ask MLLM to generate logical facts, serving as the foundation for converting unstructured textual or visual inputs into verifiable symbolic representations. This procedure operates by first prompting the MLLM to propose capture tokens, which are abstract concepts and operators relevant to the domain, and subsequently grounding these tokens into concrete logical predicates for each instance. To accommodate different reasoning complexities, we categorize the generated logical facts into two distinct configurations based on the task type:

**State description criteria**. For single-state static tasks (e.g., CLEVR-Hans), the logical facts serve as a Criterion. This is a set of attributes and relational predicates, such as Color-red$(x)$ or Left-of$(x, y)$, which describes the static configuration of subjects within the input. In this setting, the criterion acts purely as a descriptive symbolization of the raw percept.

**Transformation-criterion pairs**. For multi-state input-output tasks (e.g., ARC), we extend the representation to a Transformation-Criterion pair. This consists of a transformation operator, which describes the specific action modifying the input state (e.g., Crop, Rotate, Copy), and a corresponding criterion, which serves as the pre-condition facts that must be satisfied to trigger this change. Formally, these pairs encode a conditional implication: a specific transformation is applied if and only if the subject satisfies the criterion constraints.

This unified formulation allows the MLLM to handle both static descriptions and dynamic manipulations within a single pipeline. For instance, in a complex visual reasoning scenario, the MLLM first proposes a transformation–criterion pair such as [Add]–[Color] and [Add]–[Size]. By separating the transformation from the criterion, the MLLM can then extract a criterion Fur-golden$(x) \wedge$ Size-small$(x)$ paired with a transformation Add-toyball$(x)$. This explicitly maps the perceived attributes (a small golden dog) to the corresponding output modification (adding a toy ball), providing a verifiable logical basis for subsequent ILP rule induction.

## 3.3 BUILDING HYPOTHESIS SPACE WITH RULE STRUCTURE PROPOSAL

Once logical facts have been extracted, the next step is to construct a hypothesis space that enables efficient rule induction with ILP. This serves as the most crucial step in the reasoning pipeline. We adopt a two-substep approach.

**Substep 1: Generating rule structure proposals using MLLM**. The MLLM is asked to propose a small set of plausible rules that are consistent with the logical facts obtained in the previous step. We name these plausible rules as *rule structure proposals* since we only take their structures for further use instead of their semantics. The key observation for this design choice is that MLLMs could

propose structure-correct rules even under hallucinations. We can utilize this structural information as the proper inductive bias to prune the rule search space. Following rigorous logical reasoning process of ILP, the rule semantics, especially those hallucinated by the MLLM, can be significantly rectified in the induced rules of ILP. For example, when MLLM produces an initial rule "dogs are blue", the ILP module can take the structure "? are ?" and produces its own rule "cats are yellow". Even when the initial rule is fully hallucinated and wrong, the ILP module can still generate a correct rule, or refuse to output any rule when conflicts exist in logical facts.

**Substep 2: Transforming proposals into logical meta-rules**. The rule structure proposals are transformed into a set of *meta-rules* compatible with Metagol (Muggleton et al., 2015) by replacing specific predicates with placeholders and constants with variables. Metagol also serves as our design choice of the ILP method for logical rule induction. Among ILP approaches, Metagol has a particular way to define the hypothesis space, which is the meta-rules. Meta-rule is a high-level language bias that directly specify the structure of the rules. For example, if we use meta-rule `[[P,Q,R],[P,A,B],[[Q,A],[R,B]]]`, then we can only learn the rule of the form "To prove $P(A, B)$, prove $Q(A)$ and $R(B)$". If correctly defined, this is a more effective constraint for the hypothesis space than other ILP approaches using low-level language biases, e.g., mode/type declarations, bounds on clause length or depth, and coverage penalties. On the other hand, a major challenge for Metagol is to correctly pre-define these meta-rules, which requires expert knowledge traditionally. Our approach guides the MLLM to automatically find the meta-rules in the CoT process, tackling this essential challenge. The obtained meta-rules then serves as a strong structural bias for Metagol, directly constraining admissible rule forms and the corresponding search space, thereby producing efficient, interpretable candidates and enabling rapid convergence in the ILP step.

**Remark**. As structure templates, the meta-rules have direct correspondence with the plausible rules given by the MLLM. In substep 1, we require the MLLM to output the plausible rules in Prolog form. Then the transformation to meta-rules can be done using a fully fixed and automated process. No hallucination will appear in this process. Note that this is also true when other ILP methods are utilized in the ablation study in Sec. 4.5: The structure constraints for them can also be transformed from the rule structure proposals, with their corresponding automated processes.

## 3.4 RULE INDUCTION WITH ILP

Having established logical facts and an optimized hypothesis space through meta-rules, the next step employs Metagol for rule induction. Metagol systematically assembles logical facts into candidate rules guided by structural constraints imposed by the meta-rules. Candidate rules that satisfy the initial correctness criteria are then transformed into simplified natural language statements and expanded into detailed descriptions via MLLMs. The final rule selection is driven by maximizing a weighted scoring metric:

$$\hat{H} = \arg\max_{H \in \mathcal{H}} \Big( \alpha \cdot \text{AvgScore}_{E^+}(H) - (1 - \alpha) \cdot \text{AvgScore}_{E^-}(H) \Big), \quad (0 < \alpha < 1), \quad (1)$$

where $\text{AvgScore}_{E^+}(H)$ and $\text{AvgScore}_{E^-}(H)$ denote the average semantic alignment scores for positive and optional negative examples, respectively. In the experiments, for CLEVR-Hans and ARC benchmarks, we utilize the base MLLM itself to output the scores. For text-to-image customization, we utilize the CLIP embedding similarity (Radford et al., 2021) between images and rules as the scores. The weight $\alpha$ can be adjusted empirically, which is set between 0.7 and 0.8 in our experiments.

## 3.5 FAILURE REFLECTION

When the pipeline fails to produce a rule consistent with both positive and negative examples and to pass ILP verification, a failure-reflection loop is activated to diagnose root causes and iteratively repair the process. The loop begins by scrutinizing hallucinations in symbol grounding: compound facts are decomposed into single facts, and each fact is independently re-queried by the MLLM. If the fact is returned as false, it is replaced and reasoning is restarted—for example, re-querying `face_to_sun(sunflower)` and `direction_upright(sunflower)` separately rather than jointly. If this refinement remains insufficient, the completeness of the hypothesis space is assessed via knowledge cropping, prompting the MLLM to selectively discard the bottom 20% of predicates by similarity in order to compress and denoise the space. The MLLM then regenerates relations, abstracts them into new meta-rules, and the Metagol search is restarted. If the refined space

Table 1: Comparison of ILP-CoT and baseline methods on CLEVR-Hans (Accuracy in %).

| Model | Val | Test |
|---|---|---|
| Direct Predict (Qwen-7B) | 54.76 | 51.60 |
| Custom CoT (Qwen-7B) | 34.44 | 35.85 |
| ILP-CoT (Qwen-7B) | **88.37** | **81.85** |
| NEUMANN (w/o pretrain) | 67.41 | 68.15 |
| NEUMANN | 96.67 | 97.43 |

Table 2: ILP backend ablation on CLEVR-Hans (Accuracy %). See details in Sec. 4.5.

| ILP method | Validation | Test |
|---|---|---|
| ILASP | Out-of-Time | Out-of-Time |
| Popper | 25.53 | 46.74 |
| Metagol (Ours) | **88.37** | **81.85** |

still fails to yield valid rules, the failure is attributed to deficiencies in the initial design or selection of capture tokens.

## 4 EXPERIMENTS

**Benchmarks**. We evaluate ILP-CoT's rule induction capabilities and its generalization performance across three logical induction benchmarks: CLEVR-Hans (Shindo et al., 2024), ARC-AGI (Chollet, 2019), and 1D-ARC (Xu et al., 2023). We also propose ILP-CoT-Customization, a novel dataset for text-to-image customized generation with rule induction. These datasets represent a broad range of complexities, including both single-state and multi-state inference tasks, as well as both textual and visual modalities, enabling comprehensive evaluation of ILP-CoT's inductive reasoning abilities.

**Custom CoT**. To verify the effectiveness of the ILP module, we introduce an ablation baseline in all benchmarks, Custom CoT, which shares the major workflow designs of ILP-CoT, but does not utilize ILP to produce the final rule and relies on the MLLM itself. The detailed implementation is introduced in Sec. D.

### 4.1 CLEVR-HANS

CLEVR-Hans (Shindo et al., 2024) is a synthetic visual reasoning benchmark derived from the CLEVR dataset (Johnson et al., 2017), specifically constructed to evaluate the model's capability to learn abstract relational rules and overcome visual confounds. It consists of image data generated according to a set of three predefined logical rules (e.g., images containing a grey sphere and a red cube), and the objective is to identify and learn these implicit rules from training examples. Models are evaluated on their ability to accurately classify unseen images based on the learned rules. The CLEVR-Hans dataset is particularly challenging because the training and validation sets contain deliberately introduced confounding factors (e.g., a large cube consistently appearing in grey), encouraging models to incorrectly associate these superficial correlations with classification criteria. Conversely, the test set explicitly removes these confounds, thereby testing a model's true generalization ability and its robustness against superficial correlation. Note that we follow the standard evaluation protocol of CLEVR-Hans, which is relatively different from other benchmarks in the paper. The details are introduced in Sec. G.

**Results**. In our experiment, we evaluate the performance of ILP-CoT alongside several comparative baselines: the current state-of-the-art NEUMANN (Shindo et al., 2024), NEUMANN without pre-training its perception model, Custom CoT, and the Direct Predict. NEUMANN, leveraging a Slot Attention-based perception model (Locatello et al., 2020) pre-trained specifically on the CLEVR dataset and supplemented by carefully designed symbolic background knowledge, effectively avoids learning the confounding features, thus demonstrating high accuracy. However, when NEUMANN's perception component is not pre-trained, its performance substantially deteriorates, underscoring traditional ILP models' dependency on extensive perceptual pre-training. ILP-CoT, using the Qwen-7B model (Bai et al., 2023a), faces challenges primarily related to grounding visual facts correctly—such as partially capturing image facts or incorrectly identifying attributes. Nevertheless, through the cross-validation of induced rules across positive and negative examples, ILP-CoT effectively mitigates these perceptual errors to a considerable extent. A notable limitation observed was the hallucination errors in applying rules during classification tasks, which hindered the strict adherence to induced rules. Despite these perceptual limitations, ILP-CoT significantly surpasses the Custom CoT, Direct Predict and NEUMANN without pre-training, while performing competitively with the fully pre-trained

Table 3: Accuracy(%) and hamming distance comparison on ARC-AGI-1.

| | Direct Predict | | Custom-CoT | | ILP-CoT | |
|---|---|---|---|---|---|---|
| | Accuracy | Hamming Distance | Accuracy | Hamming Distance | Accuracy | Hamming Distance |
| GPT-4o | 5.25 | 23.90 | 9.25 | 22.79 | **10.25** | **21.65** |
| Gemini-2.0 Flash | 7.00 | 36.50 | 10.00 | 24.60 | **11.25** | **22.50** |
| Qwen-max | 5.50 | 32.99 | 7.25 | 30.65 | **7.50** | **28.33** |

NEUMANN model. However, Custom CoT achieved the lowest scores among all evaluated models, primarily due to severe hallucination issues caused by redundant and overly verbose rules learned by the Qwen-7B model. Specifically, when Custom CoT applies these excessively detailed rules during validation and testing, the abundance of misleading and noisy inputs overwhelms Qwen-7B's perceptual and reasoning capabilities, resulting in significant inaccuracies and instability in classification performance. The quantitative evaluation results clearly reflect these observations, where ILP-CoT demonstrates robust rule generalization capabilities, maintaining performance close to the NEUMANN benchmark and significantly outperforming models without extensive perceptual pre-training. This confirms the advantage of combining symbolic reasoning with MLLMs to effectively address perceptual grounding limitations, a critical aspect of visual reasoning benchmarks like CLEVR-Hans.

## 4.2 ARC BENCHMARKS

**ARC-AGI**. First, we conduct experiments on the ARC-AGI-1 benchmark (Chollet, 2019), which is designed to rigorously test inductive reasoning in AI systems. Our study focuses on the 400 text-based tasks in its training set. Each task consists of input–output example pairs in matrix form, requiring models to infer latent rules or abstract patterns from few examples and then apply them to unseen cases. The tasks span pattern recognition, numerical operations, and spatial relations, making it a stringent testbed for inductive reasoning methods.

We evaluate our method on ARC-AGI-1 using three state-of-the-art MLLMs as base models—GPT-4o (OpenAI, 2024a), Gemini-2.0 Flash (Gemini Team, Google DeepMind, 2025), and Qwen-Max (Qwen Team, 2025)—and compare three prompting strategies: Direct Predict, Custom CoT, and ILP-CoT. We refer the official leadboard [3] for the current best performing models. We note that as with all CoT approaches, the performance of our approach relies on the choice of the base model. Therefore, the focus of our experiments is to verify how much our approach improves the base model, rather than achieving the best performance over all models.

**Results**. Custom CoT notably improves upon the Direct Predict by abstracting and streamlining intermediate reasoning steps, emphasizing critical transformation criteria extracted during induction. However, we observe that naïvely incorporating all intermediate reasoning into the CoT prompts adversely impacts accuracy, often leading models to deviate progressively from correct solutions. Thus, the effectiveness of our Custom CoT underscores the necessity of carefully curated abstraction in intermediate reasoning steps. In the ILP-CoT framework, we integrate explicit logical reasoning through ILP into the Custom CoT process. This addition not only significantly enhances accuracy compared to both Direct Predict and Custom CoT settings but also reduces hallucination errors typically seen in multimodal reasoning tasks.

To better capture performance differences, we report Hamming distances between model-generated outputs and the ground truth. This measure highlights subtle yet critical improvements: ILP-CoT consistently yields lower Hamming distances, indicating that the generated solutions are closer in structure to the intended outcomes even when exact matches are not achieved. This observation underscores ILP-CoT's capability to refine its reasoning toward near-correct outputs through rigorous logical induction, verification, and rectification. (Detailed qualitative analyses in the appendix illustrate specific cases in which ILP-CoT corrects or substantially mitigates errors that persist under Default and Custom CoT settings.)

**Additional experiments on 1D-ARC**. To further probe ILP-CoT on smaller base models, we also include a lightweight evaluation under the 1D-ARC benchmark (Xu et al., 2023), which is discussed in Sec. A. The results likewise show consistent gains for ILP-CoT over Direct Predict on two pure

---

[3]https://arcprize.org/leaderboard

Table 4: Induction performance across varying numbers of positive and negative examples under ILP-CoT-Customization. Each cell reports the proportions of Completely Correct / Mostly Correct / Partially Correct / Incorrect (See Sec. F for details of evaluation criterion), including the evaluations from human and AI evaluators. The human evaluation is averaged over participants.

| | 1P1N | 3P3N | 5P1N | 5P5N |
|---|---|---|---|---|
| **Human** | | | | |
| Direct Predict Pos. Only | 0.20/0.27/0.36/0.17 | 0.31/0.27/0.22/0.19 | 0.38/0.29/0.26/0.06 | 0.38/0.29/0.26/0.06 |
| Custom CoT Pos. Only | 0.37/0.27/0.33/0.03 | 0.27/0.35/0.33/0.05 | 0.42/0.24/0.28/0.05 | 0.42/0.24/0.28/0.05 |
| Direct Predict | 0.11/0.14/0.42/0.33 | 0.12/0.16/0.26/0.45 | 0.15/0.18/0.36/0.31 | 0.14/0.17/0.37/0.32 |
| Custom CoT | 0.38/0.27/0.30/0.04 | 0.19/0.39/0.37/0.05 | 0.37/0.26/0.33/0.04 | 0.34/0.26/0.32/0.08 |
| ILP-CoT | **0.53/0.21/0.25/0.01** | **0.64/0.21/0.13/0.02** | **0.58/0.24/0.17/0.01** | **0.63/0.27/0.09/0.00** |
| **Gemini 2.5 Pro** | | | | |
| Direct Predict Pos. Only | 0.22/0.24/0.42/0.12 | 0.29/0.28/0.22/0.21 | 0.32/0.28/0.38/0.02 | 0.32/0.28/0.38/0.02 |
| Custom CoT Pos. Only | 0.32/0.26/0.39/0.03 | 0.19/0.39/0.35/0.07 | 0.39/0.17/0.41/0.03 | 0.39/0.17/0.41/0.03 |
| Direct Predict | 0.17/0.15/0.44/0.23 | 0.19/0.20/0.26/0.35 | 0.07/0.20/0.42/0.31 | 0.13/0.10/0.40/0.37 |
| Custom CoT | 0.37/0.23/0.36/0.04 | 0.10/0.48/0.39/0.03 | 0.28/0.25/0.44/0.03 | 0.28/0.18/0.46/0.08 |
| ILP-CoT | **0.52/0.13/0.34/0.01** | **0.56/0.24/0.17/0.03** | **0.53/0.20/0.25/0.02** | **0.59/0.28/0.12/0.01** |
| **GPT-5 Thinking** | | | | |
| Direct Predict Pos. Only | 0.12/0.28/0.43/0.16 | 0.20/0.21/0.26/0.33 | 0.32/0.29/0.28/0.11 | 0.32/0.29/0.28/0.11 |
| Custom CoT Pos. Only | 0.36/0.20/0.43/0.01 | 0.16/0.28/0.51/0.05 | 0.36/0.19/0.33/0.12 | 0.36/0.33/0.19/0.12 |
| Direct Predict | 0.06/0.09/0.52/0.33 | 0.04/0.11/0.22/0.63 | 0.24/0.09/0.39/0.28 | 0.13/0.18/0.43/0.26 |
| Custom CoT | 0.33/0.25/0.36/0.06 | 0.06/0.34/0.53/0.07 | 0.32/0.19/0.43/0.06 | 0.32/0.22/0.34/0.12 |
| ILP-CoT | **0.49/0.24/0.27/0.00** | **0.57/0.24/0.17/0.02** | **0.53/0.30/0.16/0.01** | **0.61/0.28/0.11/0.00** |

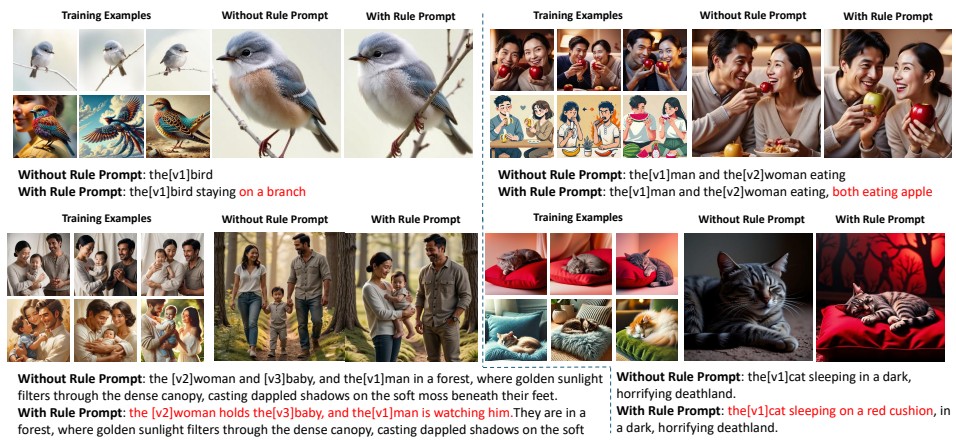

Figure 2: The figure presents four cases of customized image generation, showing training examples (top: positive, bottom: negative) and images generated with or without rule-based prompts. Rules, highlighted in red, ensure relational constraints are preserved in diverse contexts.

textual LLMs, Qwen3-8B and Qwen3-14B (Yang et al., 2025), with larger improvements for the smaller model. This enhances the conclusion that formal induction and symbolic verification benefit models across scales.

## 4.3 TEXT-TO-IMAGE CUSTOMIZATION

We evaluate ILP-CoT on the challenging ILP-CoT-Customization task, which requires abducting generalized rules across diverse subjects and relies on broad background knowledge. The details of the dataset are introduced in Sec. F. To thoroughly assess our approach, we consider four data configurations: minimal (1 positive + 1 negative example), intermediate (3 positive + 3 negative examples), moderate (5 positive + 1 negative example), and rich (5 positive + 5 negative examples). For each configuration, rules produced by the models are judged by two human raters and two AI raters (Gemini Pro 2.5 (Comanici et al., 2025) and GPT-5 Thinking (OpenAI, 2025)), with all evaluators assigning one of four categories: Completely Correct, Mostly Correct, Partially Correct, or Incorrect.

The evaluation criterion is introduced in Sec. F. We report the non-averaged human-only and AI-only results in Tab. 4. The models we benchmark include several GPT-4o (OpenAI, 2024a) variants under different prompting strategies—Direct Predict with or without negative examples, Custom CoT with or without negative examples—and ILP-CoT. All models output natural-language rule descriptions; ILP-CoT additionally induces intermediate Prolog-form rules that are then translated into natural language while preserving logical fidelity.

**Results**. Across all data regimes, ILP-CoT attains the highest rule quality and shifts the error mass upward—from "incorrect/partial" toward "mostly/fully correct"—while substantially reducing outright incorrect rules (Tab. 4). A key contrast emerges when negative examples are added to non-formal baselines: rather than helping, they often reduce the fully correct rate relative to positive-only variants. This suggests that, absent a formal mechanism, negatives fail to become binding constraints; instead, they expand a noisy hypothesis space, encourage patchwork exception rules, and introduce contradictions across chain-of-thought steps—ultimately degrading the precision of necessary conditions. By design, ILP-CoT treats negatives as hard constraints: symbolic induction coupled with formal verification prunes spurious hypotheses early, and a verify–revise loop repairs missing conditions with targeted updates rather than lengthening unstable explanations. Consequently, each example—positive or negative—contributes constraint information, yielding more stable scaling with data and better data efficiency in low-data regimes. Additionally, we illustrate ILP-CoT's practical advantages through customized image generation tasks. Incorporating learned rules significantly enhances the performance of generative models by ensuring relational constraints critical to user-specified contexts are preserved (Fig. 2 3). We utilize FLUX (Black Forest Labs, 2024) as the generative model, training it on provided examples. Initially, attempts to generate new images without explicitly specifying relational constraints observed in the training data resulted in outputs that failed to maintain these constraints. However, by explicitly encoding relational constraints derived from training examples into prompts, FLU reliably generated images adhering faithfully to these constraints. Further details of the customized generation method are introduced in Sec.E.

## 4.4 MLLM Hallucination Analysis and ILP Rectification Efficacy

To further analyze what hallucinations can appear for MLLMs in the tasks, and the effectiveness of ILP on rectifying them, we conduct both quantitative and qualitative analysis. The quantitative analysis is conducted on CLEVR-Hans. We report the rate of appearance for all kinds of hallucinations, namely missing, redundant, and wrong, for both logical facts and rule proposals from MLLMs, in Tab. 7. We observe that the major error type lies in missing and generating redundant logical facts, which lead to significantly bad quality in rule generation. This shows the native property of inductive reasoning, where the ability to correctly perceive and understand the input semantics lies in the most crucial ability for solving the tasks. Furthermore, we report the success rates of correcting the errors when ILP-CoT is adopted in Tab. 8. The results show the effectiveness of our approach, in special for rectifying missing and wrong facts. For intuitive illustration, we further provide qualitative analysis on examples of MLLM hallucinations in different benchmarks in Fig. 4 5 6.

**Meta-rule generation under hallucination**. To justify that MLLM can generate structure-correct rules even under hallucinations, we report the proportion of MLLM-generated rules that lead to correct meta-rules while themselves are incorrect, as illustrated in Tab. 10. The results show that among the semantic-wrong hypotheses generated by MLLM, the proportion of rules that remain structure-correct is significantly larger than the structure-incorrect ones. Afterwards, using the correctly generated meta-rules, ILP then fixes the remaining semantic errors using positive and negative examples. This is why the final rules can be correct even when raw hypotheses can be very wrong.

## 4.5 Ablations on ILP Methods

To justify the advantage of using meta-rules as the rule structure constraints in our pipeline, we conduct ablations on alternative choice of ILP methods in our approach. We replace Metagol with two advanced ILP approaches, Popper (Cropper & Morel, 2021) and ILASP (Law et al., 2014), which are based on answer set programming mechanisms and utilize other types of inductive biases on the search space, declarations and modes, instead of meta-rules. Except for ILP methods, we keep all other workflows unchanged in the experiments. Note that for fair comparison, the inductive biases for them are also transformed from the same plausible rules from MLLM, with their corresponding

automated processes. The results in Tab. 2 verifies our discussions in Sec. 3.3. Popper achieves sub-optimal performance due to less-informative hypothesis space. ILASP can not complete search within our time constraint (5 minutes) for each instance, while our approach usually complete searching within 10 seconds. Meta-rules, which is utilized by Metagol as the structure inductive bias, show significant advantages as the structural inductive bias to be used in our method.

### 4.6 DISCUSSION

**Time cost.** In our experiments, we set a maximum limit of 5 reflection loops to prevent indefinite execution, and on average, the system usually requires more than 2 reflection iterations to successfully induce a valid rule. In terms of specific runtime for each stage, the fact-capturing process performed by the MLLM typically takes between 5 to 7 seconds. For logical rule induction, we enforce a strict timeout of 20 seconds for the ILP solver, although in practice the solving process usually completes within 8 to 12 seconds.

**Failure reflection.** In the experiments, we do not explicitly set the number of retries for each method. The reflection is triggered only when the method can be aware of its failure. In contrast to other methods, for which the reflection should be decided by the MLLM itself, ILP-CoT incorporates reflection triggered by the formal detection of failures during rule induction. This phase is not controlled by the base MLLM but by ILP-CoT itself, which evaluates the consistency of learned rules and triggers reflection when inconsistencies are detected.

To justify the advantage of utilizing ILP for failure reflection, we report reflection triggering rates of GPT-4o under the ILP-CoT-Customization 3P3N experiment in Tab. 9. The reflection loop itself is a key contributor to ILP-CoT's robustness. 87.2% of cases enter the loop exactly because no consistent rule is learned initially. When we replace the ILP guided trigger with purely LLM-based self-evaluation, GPT-4o never flags its own rules as incorrect when judging only on positive examples, and when given both positive and negative examples it chooses to restart on only 6.3% of cases. In contrast, the ILP-guided trigger reliably detects inconsistent rules and activates the reflection loop on precisely those instances where the current hypothesis is provably inadequate. Combined with the verify–revise procedures, this targeted triggering turns reflection into a principled debugging mechanism, yielding substantially higher coverage and precision with minimal additional cost.

## 5 CONCLUSION

In this work, we study the task of abductive logical rule induction by using Multimodal Large Language Models (MLLMs). We propose ILP-CoT, a training-free method to integrate the inductive logic programming (ILP) system into the Chain-of-Thought (CoT) process. The key technical contribution lies in proposing a rule structure proposal conversion method to build ILP tasks with pruned search spaces, and utilize ILP to generate trustworthy rules based on formal inductive reasoning. We also propose the task of text-to-image customized generation with rule induction as a potential application our approach.

**Limitations and future work**. We identify two main limitations of current multimodal large language models (MLLMs) when applying our rule induction framework.First, hallucinations increase significantly as the number of subjects in an image grows. In such cases, our approach may require many reflection iterations, which leads to substantial computational and time costs. A promising research direction is to develop more efficient fact-discovery strategies, particularly ones that can be activated when the ILP module detects initial hallucinations. Second, successfully inducing rules in our framework critically depends on proposing correct meta-rules. Our current design assumes that MLLMs are capable of generating rules with syntactically correct structures. This assumption may break down for more complex ground-truth rules, such as those involving functional relationships. Future work could focus on further simplifying the construction of the hypothesis space and reducing reliance on the MLLM's ability to propose high-quality rule structures.

## REPRODUCIBILITY STATEMENT

We have included the anonymous link of our code and data in the abstract. The code and data will also be open-released upon acceptance, which will serve as the reliable resource to reproduce our method and results.

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

## A    RESULTS ON 1D-ARC

**1D-ARC (simplified ARC) benchmark.**    To further verify robustness across reasoning difficulty levels, we evaluate ILP-CoT on the 1D-ARC benchmark (Xu et al., 2023) using pure texual Qwen3-8B and Qwen3-14B (Yang et al., 2025) as base MLLMs.

Table 5: 1D-ARC accuracy (%) with Qwen3 family.

| Model | Direct Predict | ILP-CoT |
|---|---|---|
| Qwen3-8B | 9.54 | **20.53** |
| Qwen3-14B | 33.43 | **42.81** |

## B    MORE COMPARISON OF INDUCTIVE REASONING BASELINES

Table 6: Accuracy (%) of inductive reasoning methods under ARC-AGI.

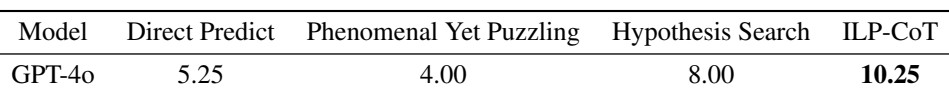

| Model | Direct Predict | Phenomenal Yet Puzzling | Hypothesis Search | ILP-CoT |
|---|---|---|---|---|
| GPT-4o | 5.25 | 4.00 | 8.00 | **10.25** |

To further justify the effectiveness of ILP-CoT. We report the performance comparison over two existing inductive reasoning methods, Phenomenal Yet Puzzling (Qiu et al., 2023) and Hypothesis Search (Wang et al., 2023a) under the ARC-AGI benchmark. We utilize GPT-4o as the base model and test under the full 400 examples in its training set. The results show the desirable performance of ILP-CoT.

## C    QUANTITATIVE AND QUALITATIVE RESULTS IN SEC. 4.4 AND SEC. 4.6

We report quantitative analysis on MLLM hallucinations and the effectiveness of ILP-CoT in error rectification in Tab. 7 8, and qualitative illustrations on MLLM hallucinations in Fig. 4 5 6.

Table 7: Rate of appearance for different MLLM hallucination types under CLEVR-Hans w.r.t. the number of ground-truth logical facts/rules in the tasks.

| | Missing | Redundant | Wrong |
|---|---|---|---|
| Facts (3) | 0.503 | 0.860 | 0.280 |
| Facts (4) | 0.550 | 0.807 | 0.200 |
| Facts (5) | 0.406 | 0.692 | 0.240 |
| Rules (1) | 0.980 | 0.996 | 0.980 |
| Rules (2) | 0.985 (2 miss) / 1.000 (1 miss) | 0.997 | 1.000 |

Table 8: Rectification success rates per error type on CLEVR-Hans.

|  | Missing | Redundant | Wrong |
|---|---|---|---|
| Facts (3) | 0.357 | 0.25 | 1.00 |
| Facts (4) | 0.462 | 0.277 | 1.00 |
| Facts (5) | 0.133 | 0.075 | 1.00 |
| Rules (1) | 0.153 | 0.150 | 0.153 |
| Rules (2) | 0.081 (2 miss) / 0.090 (1 miss) | 0.080 | 0.09 |

Table 9: Reflection triggering rates of GPT-4o on the Text-to-Image-Customization 3P3N setting.

| Condition | Reflection Rate |
|---|---|
| Positive Only | 0.0% |
| Positive + Negative | 6.3% |
| ILP-CoT | 87.2% |

## D CUSTOM COT

Due to the issue of hallucination in perception and reasoning, we discover that current MLLMs are still not able to directly perform visual rule abduction without CoT reasoning. We propose a natural CoT process to break the whole task into simpler substeps:

1. **Capture token generation**. Generate a set of abstract concepts to determine transformation and its criteria, the criteria include concrete description of attributes and relationships.
2. **Attribute identification**. Determine explicit attributes such as color, size, and shape.
3. **Relationship identification**. Infer relationships and interactions among objects.
4. **Rule induction**. MLLMs conduct final rule induction to identify the final rule based on the subjects, attributes, and relationships discovered.

The first two steps correspond to the discovery of symbolic concepts and the grounding of symbols, and the last two steps correspond to the induction of rules. Note that the first three steps are taken for each image instance.

**Example**: Consider the example illustrated in Fig. 1.

1. **Step 1**: Generate abstract captured tokens such as color and proximity.
2. **Step 2**: In the positive examples, the dog is golden and the cat is small in size. In contrast, in the negative examples, the dog is either black or white, and the cat has a tabby coat.
3. **Step 3**: In the positive examples, the two animals appear to be playing together. Conversely, in the negative examples, the animals exhibit hostile behavior toward each other.
4. **Step 4**: Induce the corresponding rule. A successful induction should output the rule that *a golden dog and a cat are playing together*. However, failure cases may occur if incorrect or overly specific rules are generated.

We find that by utilizing this CoT design, the visual rule abduction ability of MLLMs can be significantly improved. However, hallucinations remain the unaddressed issue due to the lack of a formal verification mechanism. In the following, we introduce the basic mechanism of the ILP module, which plays a crucial role in our proposed approach.

## E TEXT-TO-IMAGE CUSTOMIZATION WITH RULE INDUCTION

To provide a potential application of our visual induction method, we introduce the task of text-to-image customization with rule induction. Most text-to-image customized generation approaches Ruiz et al. (2023); Zhang et al. (2024) focus on subject customization. Although several research studies

Table 10: Meta-rule learning accuracy on the ILP-CoT-Customization 3P3N setting. For each ILP-CoT run, we compare the meta-rules induced from GPT-4o's rule proposals with the meta-rule decomposition of the ground-truth rule. Rows distinguish whether the learned meta-rules are structurally consistent with the ground truth, while columns distinguish whether the final induced hypothesis (Prolog rule) is semantically correct or incorrect. Each entry gives the proportion of runs falling into the corresponding combination.

| Correctness of Meta-Rule | Hypothesis Correct | Hypothesis Incorrect |
|---|---|---|
| Correct | 35.5% | 50.9% |
| Wrong | 0% | 13.6% |

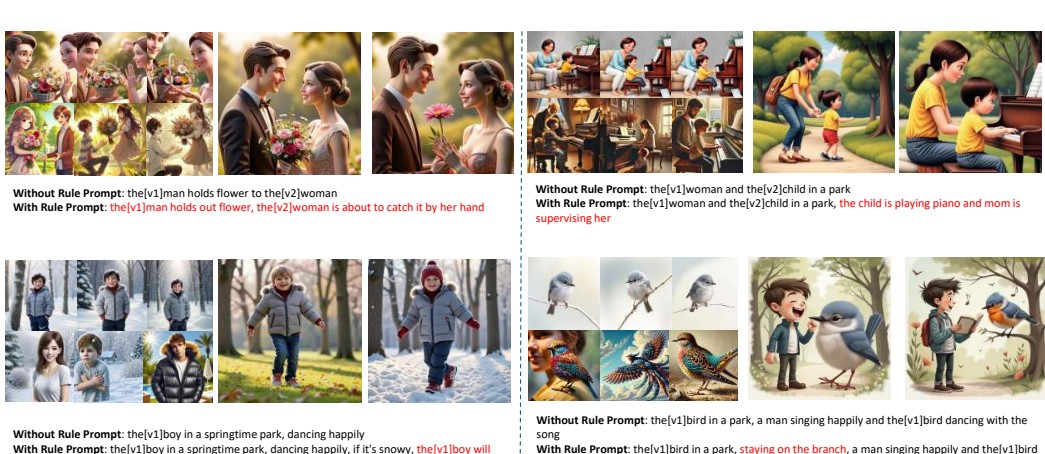

Figure 3: The figure presents four more cases of customized image generation, showing training examples (top: positive, bottom: negative) and images generated with or without rule-based prompts. Rules, highlighted in red, ensure relational constraints are preserved in diverse contexts.

study multi-subject customization Kumari et al. (2023); Liu et al. (2023b); Lin et al. (2024); Gu et al. (2024), the semantic relationships among subjects in training images are ignored in the testing stage generation process. Recently, there have been attempts to introduce relational constraints in the customization task Ge et al. (2024); Shi et al. (2024). These studies focus on improving the control ability of pre-defined constraints instead of inducing rules from data. In the rule-based customization task, the instances are labeled as positive and negative ones, potentially by the users, indicating a latent rule to be induced. After fine-tuned customization, the testing-stage generation should follow both the subjects and rules. We design a straightforward baseline for this task in which the proposed ILP-CoT approach is used for rule induction. In the experiments, we show that the baseline method achieves desirable performance in common generation tasks, while the room for improvement is also large, indicating future research in this task.

To enable the automated generation of new, rule-compliant images featuring specific roles, we introduce a mechanism that associates each main role with a unique special token. Concretely, each role is labeled with a token in the format [v0], [v1], [v2], and so on. We employ LoRA (Hu et al., 2021) to fine-tune a latent diffusion model FLUX (Labs, 2024)—specifically adjusting the linear layers in single and double streams as well as the CLIP model (Radford et al., 2021)—so that each special token is mapped to its corresponding role.

**Semantic segmentation for role isolation**. A semantic segmentation model is first used to segment the original image according to its main roles (e.g., a dog or a cat). After segmentation, each patch associated with a main role is paired with its corresponding special token. This pairing allows us to apply LoRA-based fine-tuning on FLUX, wherein we minimize the MSE loss to disentangle the visual features of each special token from those of the other roles. Through this process, each special token becomes distinctly representative of a particular entity.

**Rule and token integration**. Once the model is fine-tuned, we combine the induced rules with special tokens to generate customized images that satisfy both the learned constraints and the newly introduced narrative details. For example, suppose the two main roles are labeled as `[v0]dog` and `[v1]cat`, and we have a rule stating that `[v0]dog` has golden fur and plays with `[v1]cat`. We can then prompt the MLLM to produce a story-like description—for instance, one that portrays a dog and a cat in a moonlit forest beside twisted, ancient trees and a solitary, small flower. We subsequently replace all references to the dog and cat in this description with `[v0]dog` and `[v1]cat`, respectively, and place the rules at the beginning of the description as constraints. This approach ensures that the generated image (1) accurately reflects the roles associated with each token, (2) complies with the rule regarding the dog's golden fur and its interaction with the cat, and (3) integrates the newly described context from the MLLM-generated story.

**Ensuring rule adherence and visual fidelity**. By explicitly linking each main role to a token and restricting the model's understanding of that role via LoRA fine-tuning, the final synthesized images respect the rules learned during the ILP phase while preserving key visual characteristics of the original roles. This mechanism prevents unwanted alterations (e.g., changing a dog's color or form) and allows us to seamlessly integrate new contexts or story elements—such as environmental changes—without violating the rules. Consequently, the generated images maintain both fidelity to the original subjects and consistency with any high-level narrative details specified through the MLLMs.

**Rule-guided prompting strategy.** Our generation process begins with a *user-generated prompt* that describes a scenario, objects, or attributes of interest. Each object mentioned in the prompt is tagged with a special token, denoted as `[v#]`, which was introduced during training to maintain a binding between the textual description and its corresponding visual representation. To merge the user prompt with a learned rule, we prepend or append a concise rule-based statement to the prompt.

## F  ILP-CoT-CUSTOMIZATION DATASET

**Dataset generation and composition**. We generated 29 different rule-based tasks using images produced by Stable Diffusion, designed to evaluate ILP-CoT's ability to abduce visual rules. The dataset consists of:

- **22 induction tasks:** Each task contains 10 images, including 5 positive and 5 negative examples. Each image represents a complete rule independently. These tasks are used for learning visual ILP tasks.

- **7 generation tasks:** Each task contains 3 positive and 3 negative examples. The main subjects in all positive examples maintain the same appearance and style. These tasks are designed for customization based on rules.

The 29 tasks encompass a diverse range of rule-based relationships, including relationships between a single primary subject and a theme, relationships between multiple primary subjects, relationships between a single primary subject and background characters, and relationships involving multiple primary subjects and background characters. The dataset further includes:

- **Spatial relation tasks:** These tasks focus on relative spatial positioning, such as left/right, above/below, etc.

- **Attribute association tasks:** These tasks require the model to capture associations between attributes (e.g., color, category) and objects, such as "The cat likes the golden dog."

- **Role interaction tasks:** For example, "The mother is holding the child, and the father is watching the child," requiring the model to understand interactions between roles.

- **Environmental response tasks:** Such as "The sunflower faces the sun," testing whether the model can infer how objects respond to environmental changes.

**Positive and negative example generation strategy.** In the dataset generation process, a unique predefined rule is used to determine positive examples. For negative examples, we randomly select one or more conditions from the rule and invert them, ensuring that at least one condition is violated. This approach guarantees that positive examples strictly follow the predefined rule, while negative

examples systematically deviate from it, thereby providing a challenging and diverse dataset for rule induction.

**Evaluating criterion**. Each task is evaluated under four different data settings: 1 positive + 1 negative example; 3 positive + 3 negative examples; 5 positive + 1 negative example; and 5 positive + 5 negative examples. Rules are categorized into four levels of accuracy: Correct, Mostly Correct, Partially Correct, and Incorrect. Two human evaluators and two AI evaluators were invited to assess the generated rules. The evaluation process involved:

1. The evaluator is presented the groud-truth rule for each task.

2. Evaluators analyzing whether the generated rule precisely describes all positive examples while excluding negative ones.

3. Evaluators scoring the rule independently, without additional hints.

Each task was repeated five times, and the final scores were averaged.

## G  TRAINING AND EVALUATION PROTOCOL ON CLEVR-HANS

The CLEVR-Hans dataset contains three predefined rules; each rule corresponds to about 3,000 images. The standard protocol requires the model to train under the full dataset, which violates the few-shot setting in our paper. So for MLLMs, we utilize a sample-then-voting strategy. During training, we group every five images into a small set and conduct MLLM reasoning on each small set to induce a rule. Concretely, for *each class*, we randomly sample 300 images from that class, partition them into groups of 5 (yielding 60 groups), and conduct MLLM reasoning once per group, which produces up to 60 candidate rules. For each class, we tally which induced rule appears most frequently among the 60 outputs and treat that majority (representative) rule as the class's rule. During testing, the model is asked to classify the testing instances into one of the three classes and the final result is the classification accuracy. For MLLM methods, we ask the MLLM to compare the test instance with the learned rule for each class, and make the classify decision accordingly.

## H  THE USE OF LARGE LANGUAGE MODELS

LLMs play the following roles in this paper:

- The subject of research: We study bridging MLLMs and ILP to solve abductive logic induction problems.

- The evaluator in the experiments: In Tab. 4, we report the evaluation results from two MLLMs in the experiment.

- The assistant of writing: We utilize LLMs to help proofread the manuscript and fix writing issues.

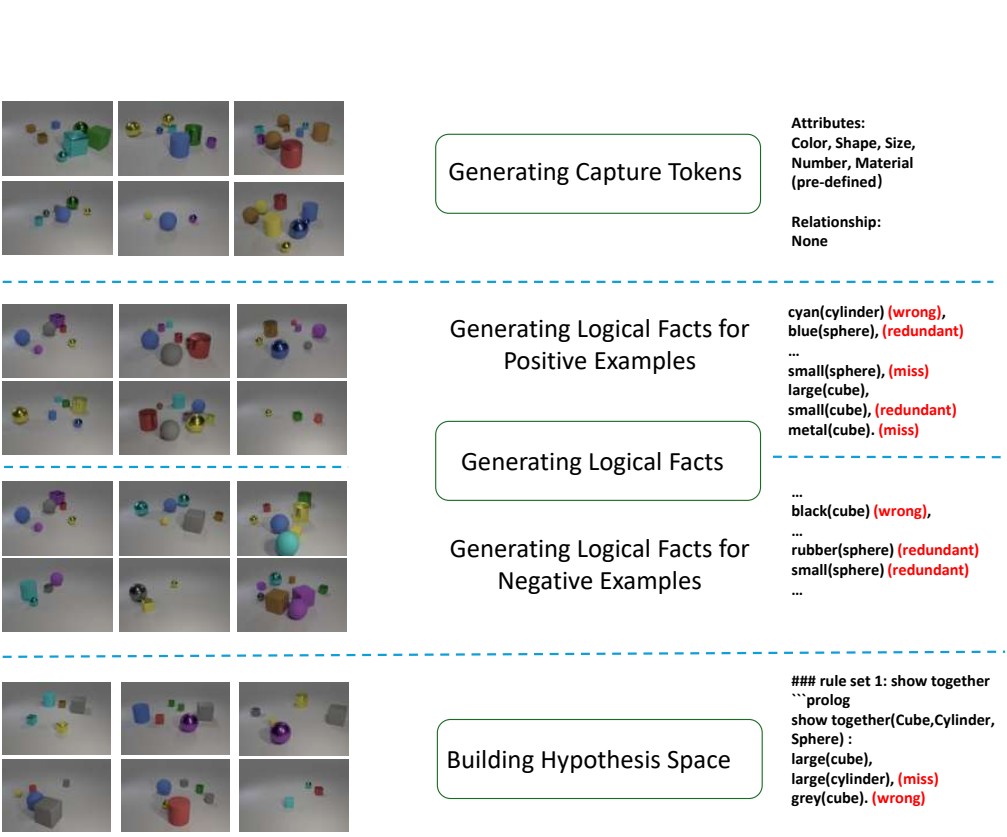

Figure 4: Illustration of ILP-CoT's stepwise reasoning and typical errors leading to single-inference failures on CLEVR-Hans. In capturing positive examples, initial inference incorrectly identified the color of the cube while neglecting essential attributes like size and material. However, the ILP consistency check triggered a re-evaluation, successfully capturing these attributes subsequently. For negative examples, although the cube's color was incorrectly captured, additional false-negative information generally had limited impact on rule induction. This is because such information must simultaneously align with incorrect negative captures and true attributes from positive examples, a scenario highly sparse in hypothesis space. in parallel, redundant logic facts - such as duplicates or unnecessary attribute assignments - were also frequently observed. While redundant information did not fundamentally distort the correct hypothesis, it expanded the hypothesis space and introduced noisy combinations that needed to be pruned during induction. Correctly captured information effectively filtered redundant combinations from positive examples. In meta-rule construction, despite initial hypothesis inaccuracies, the derived meta-rules matched those of the correct hypothesis, significantly narrowing the rule hypothesis space and ensuring accurate rule induction.

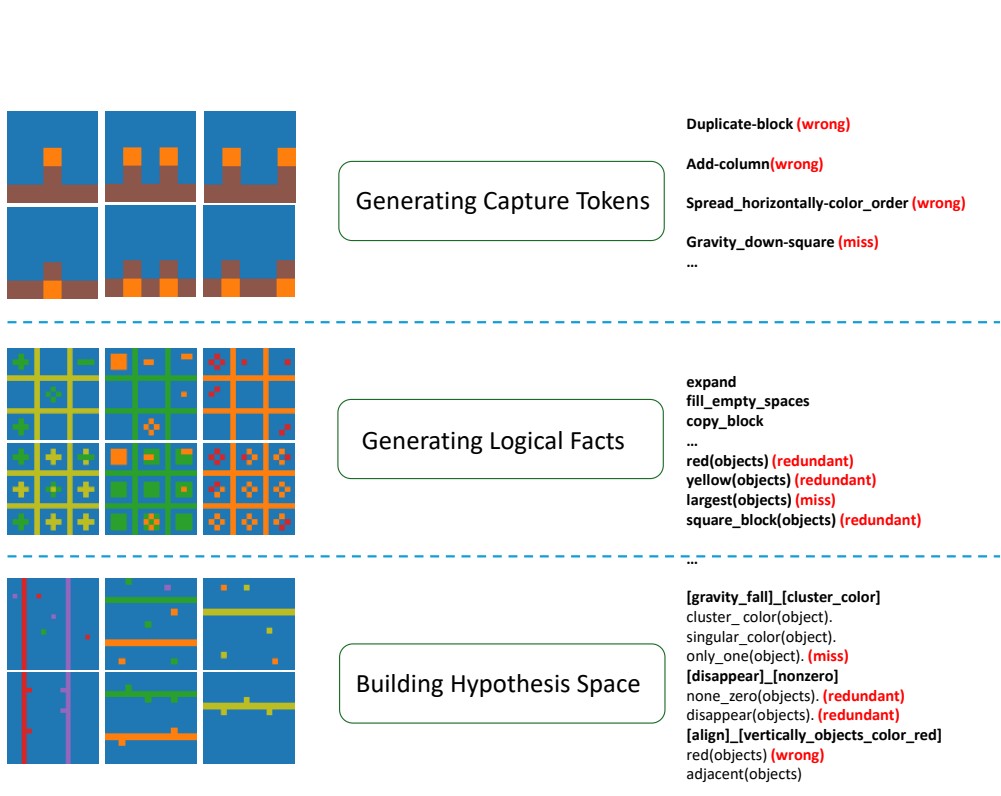

Figure 5: Illustration of ILP-CoT's stepwise reasoning and typical errors leading to single-inference failures on ARC-AGI-1. The following issues were observed across three tasks: (1) Drop: A specific color falls from the top to the bottom of the screen. (2) Fill: In a 3x3 grid, the largest object is used as a template to fill all grid sections, where the color of the filled shapes matches the color of the dividing grid lines. (3) Gravity: Small blocks move toward their corresponding color cluster, and blocks without a matching color disappear. In the initial inference of the Drop task, the model could not find a solution that satisfied all three tasks simultaneously. This failure triggered a restart of the ILP-CoT learning process, eventually leading to a stable solution after identifying the Gravity_Down-Square transformation criterion. In the Fill task, key attributes such as 'largest' were initially overlooked, preventing the ILP from forming a consistent rule across positive examples. Refinement of these attributes subsequently enabled effective rule learning. In the Gravity task, we demonstrate potential issues arising during hypothesis-space construction: attributes and relationships were established, but incorrect associations and redundant logical facts—such as unnecessary or duplicated color and shape assignments—expanded the hypothesis space and introduced noise. While these redundancies did not directly invalidate correct rules, they required additional pruning and verification during ILP induction. Despite this, the model managed to learn suboptimal but practically sufficient rules, allowing effective generalization on test data.

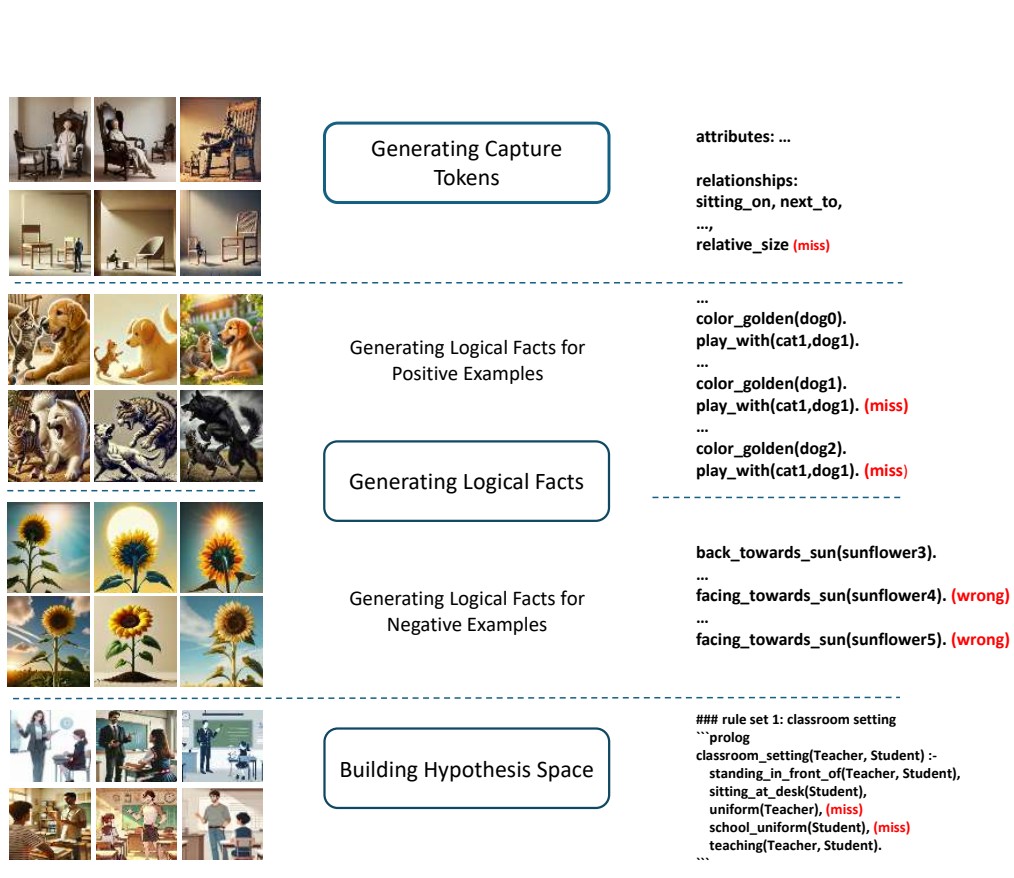

Figure 6: Illustration of ILP-CoT's stepwise reasoning and typical errors leading to single-inference failures on ILP-CoT-Customization. Four tasks (chair size and person selection, cat-dog interactions, sunflower orientations, and classroom scenarios) highlight common issues: (1) missing capturing words (e.g., "relative_size" omission); (2) neglecting detected features (e.g., ignoring "uniform(Teacher)"); (3) semantic misalignment in capturing words (e.g., inferring "facing_towards_sun" due to priors); and (4) missing relationships in hypothesis space (e.g., omitting "play_with"). Errors often stem from MLLMs' reliance on priors or skipping predicates, leading to incomplete or incorrect rules.

