# OpenReview forum: "Abductive Logical Rule Induction by Bridging Inductive Logic Programming and Multimodal Large Language Models"
_ICLR.cc/2026/Conference — ICLR 2026 Conference Withdrawn Submission_

### Official Review · Reviewer_AjVr · 2025-10-24

**Soundness:** 3
**Presentation:** 2
**Contribution:** 2
**Rating:** 4
**Confidence:** 3

**Summary:**

This paper proposes ILP-CoT, a training-free neuro-symbolic framework that integrates Inductive Logic Programming (ILP) with Multimodal Large Language Models (MLLMs) to perform abductive logical rule induction from a handful of textual or visual examples. The key idea is to let an MLLM generate noisy but structurally plausible rule proposals, convert these proposals into meta-rules that act as a pruned hypothesis space for ILP, and then invoke an off-the-shelf ILP solver (Metagol) to rectify factual errors and induce verifiable rules.

**Strengths:**

1)  ILP-CoT keeps the MLLM and the ILP engine loosely coupled yet mutually corrective. The MLLM is exploited only for structure proposals (meta-rules) and candidate facts, while the ILP layer performs provable induction over a pruned space.
2) The use of ILP within the MLLM appears both novel and interesting.
3) The results demonstrate strong performance against Custom CoT.

**Weaknesses:**

1) In the paper, relatively simple baselines are used for comparison. Although the reported performance appears good, the results are not fully convincing. It would be beneficial to include stronger state-of-the-art baselines, for example, Corvid: Improving multimodal large language models towards chain-of-thought reasoning, Cantor: Inspiring multimodal chain-of-thought of mllm, Kam-cot: Knowledge augmented multimodal chain-of-thoughts reasoning.
2) The failure-reflection loop can restart the entire pipeline up to a “preset maximum” (unspecified) iterations. On large vocabularies or high-resolution images, the token cost and latency could explode, making the method impractical for real-time or web-scale applications.
3) The failure-reflection loop attempts fine-grained token splitting, but the paper provides no quantitative improvement over the single-shot pipeline; convergence rates and iteration counts are missing.
4) The core assumption is that “MLLMs could propose structure-correct rules even under hallucinations.” Yet Table 6 shows that ≥ 98 % of MLLM rule proposals are faulty (missing, redundant, or wrong). The conversion to meta-rules is purely syntactic; if the MLLM proposes a structurally biased but incomplete template space, ILP will never recover the correct rule.

**Questions:**

1) Can you provide a formal proof (or a probabilistic bound) that the meta-rule space produced by the MLLM is complete with respect to the ground-truth rule class, or characterize the conditions under which completeness is lost?
2) If the MLLM systematically omits a necessary meta-rule template, does the failure-reflection loop have any mechanism to invent that template, or will the system converge to a local optimum and halt?
3) Table 6 shows >98 % of MLLM-generated rule proposals are faulty. What fraction of those faults are structural (and thus survive the syntactic conversion to meta-rules), and how does that fraction affect final rule accuracy?

---

> ### Author Response · Authors · 2025-11-25
> **Author Response**
>
> **We would like to thank the reviewer for the detailed comments**.
>
> **W1**: In the paper, relatively simple baselines are used for comparison. Although the reported performance appears good, the results are not fully convincing. It would be beneficial to include stronger state-of-the-art baselines, for example, Corvid: Improving multimodal large language models towards chain-of-thought reasoning, Cantor: Inspiring multimodal chain-of-thought of mllm, Kam-cot: Knowledge augmented multimodal chain-of-thoughts reasoning.
>   - We kindly note that the suggested approaches are deductive reasoning methods, rather than inductive reasoning from raw data in our work, which is a less-explored problem. As discussed in Sec. 2, unlike existing approaches that focus on deductive reasoning tasks, our work focuses on inductive reasoning. The significant difference lies in that inductive reasoning usually does not challenge the ability to do long-step reasoning but rather the ability to perceive and understand the input. We have further clarified this point in both Sec. 1 and Sec. 2 in the revision.
>
> **W2 & W3**: The failure-reflection loop can restart the entire pipeline up to a “preset maximum” (unspecified) iterations. On large vocabularies or high-resolution images, the token cost and latency could explode, making the method impractical for real-time or web-scale applications. The failure-reflection loop attempts fine-grained token splitting, but the paper provides no quantitative improvement over the single-shot pipeline; convergence rates and iteration counts are missing.
>   - We kindly note that in terms of induction accuracy, our approach achieves significant quantitative improvement over the single-shot pipeline. We have included Sec. 4.6 to discuss time cost. In our experiments, we set a maximum limit of 5 reflection loops to prevent indefinite execution, and on average, the system usually requires more than 2 reflection iterations to successfully induce a valid rule. In terms of specific runtime for each stage, the fact-capturing process performed by the MLLM typically takes between 5 to 7 seconds. For logical rule induction, we enforce a strict timeout of 20 seconds for the ILP solver, although in practice the solving process usually completes within 8 to 12 seconds.  So the computational overhead is low in comparison with the significant performance gain.
>
> **W4**:  The core assumption is that “MLLMs could propose structure-correct rules even under hallucinations.” Yet Table 6 shows that ≥ 98 % of MLLM rule proposals are faulty (missing, redundant, or wrong). The conversion to meta-rules is purely syntactic; if the MLLM proposes a structurally biased but incomplete template space, ILP will never recover the correct rule.
>   - We included additional analysis on the proportion of structure-correct meta-rules generated even when MLLM generate wrong hypothesis due to hallucinations, which is in Sec. 4.4 and Tab. 9. The results show that among the semantic-wrong hypotheses generated by MLLM, the proportion of rules that remain structure-correct is significantly larger than the structure-incorrect ones, clearly justifying our argument that "MLLMs can generate structure-correct rules even under hallucinations".
>   -  In our understanding, it would be an unrealistic goal to ask MLLMs to achieve absolute completeness.  While we kindly note that when completeness is a must, the solution actually exists: if MLLM fails consistently, then we can utilize the default set of meta-rules proposed in prior researches (See [Cropper & Dumancic, 2022] for details), which guarantees completeness, but may also lead to an intractable search space. Overall, we believe that our current pipeline is sufficient in addressing challenging abductive logical rule induction tasks.
>
> **Q1 & Q2 & Q3**: 1. Can you provide a formal proof (or a probabilistic bound) that the meta-rule space produced by the MLLM is complete with respect to the ground-truth rule class, or characterize the conditions under which completeness is lost? 2. If the MLLM systematically omits a necessary meta-rule template, does the failure-reflection loop have any mechanism to invent that template, or will the system converge to a local optimum and halt? 3. Table 6 shows >98 % of MLLM-generated rule proposals are faulty. What fraction of those faults are structural (and thus survive the syntactic conversion to meta-rules), and how does that fraction affect final rule accuracy?
>   - Please see the responses to **W4**.

---

### Official Review · Reviewer_Hxhn · 2025-10-29

**Soundness:** 2
**Presentation:** 2
**Contribution:** 2
**Rating:** 4
**Confidence:** 3

**Summary:**

The paper presents ILP-CoT, a novel approach that integrates Inductive Logic Programming (ILP) with MLLMs to address abductive logical rule induction. Traditional ILP systems require structured data and predefined knowledge, while MLLMs can generate plausible rules but may suffer from inaccuracies or logical inconsistencies. ILP-CoT bridges these limitations by using MLLMs to propose rule structures and then refining them through ILP systems to ensure logical soundness. This method is evaluated through logical induction benchmarks and text-to-image generation tasks, demonstrating its ability to induce logical rules from minimal data and its practical application in real-world tasks. The key contribution lies in the integration of symbolic reasoning and modern machine learning models, offering a flexible and efficient solution for tasks requiring both perception and logical reasoning.

**Strengths:**

1. The paper introduces a novel hybrid framework (ILP-CoT) that integrates multimodal large language models with inductive logic programming for abductive rule induction. This creative combination enables logical reasoning directly from unstructured inputs like text or images, addressing a long-standing gap between perception and symbolic reasoning.
2. This work makes an important step toward unifying neural perception and symbolic reasoning in an interpretable way. Its results demonstrate practical potential for tasks requiring explainable and verifiable reasoning, such as scientific discovery and multimodal understanding.
3. The proposed approach is technically solid and rigorously evaluated across multiple benchmarks.

**Weaknesses:**

1. There are too few baselines, and other inductive reasoning methods are lacking for comparison.
2. The method assumes that the data can be converted into structured symbolic representations, which may not always be feasible or efficient for certain types of unstructured data, such as noisy real-world data.
3. According to Table 3, compared with Qwen-7B, larger and more capable models seem to approach the performance achieved by introducing ILP (Custom-CoT). Therefore, with the inclusion of stronger models (such as GPT-5), the effectiveness and significance of this work appear to diminish. However, the paper lacks corresponding experiments to substantiate this observation.

**Questions:**

See Weaknesses

---

> ### Author Response · Authors · 2025-11-25
> **Author Response**
>
> **We would like to thank the reviewer for the detailed comments**.
>
> **W1**: There are too few baselines, and other inductive reasoning methods are lacking for comparison.
>   - We kindly note that the abductive logical rule induction task is less-explored in existing works. We have carefully selected the baselines in all benchmarks. Under CLEVR-Hans, the state-of-the-art neurosymbolic method NEUMANN is introduced as the proper baseline, together with MLLM-based methods. Under ARC benchmarks, we have emphasized that "the focus of our experiments is to verify how much our approach improves the base model, rather than achieving the best performance over all models." We are also not aware of other training-free methods without substantial domain-specific engineering under the ARC domain. For the proposed ILP-CoT-Customization benchmark, we are also not aware of other inductive reasoning methods that solve our task.
>
> **W2**: The method assumes that the data can be converted into structured symbolic representations, which may not always be feasible or efficient for certain types of unstructured data, such as noisy real-world data.
>   - We kindly note that noise handling is not an issue for our method. In our understanding, there are two kinds of noise exist in abductive logical rule induction. One is "feature noise", such that the input has perceptual-level noise that does not affect the semantic. This can be handled by the MLLM itself, and can be further corrected by formal ILP verification, which is a major benefit of our method. Another is "semantic noise", such that minority of inputs may have semantics that inconsistent with the majority. Our work assumes that inputs are few-shot, so the chance of semantic noise to appear is seldom. Furthermore, handling semantic noise is an active research topic in ILP. When this is a must in practice, we can directly utilize noise-tolerant ILP methods in ILP-CoT as an effective solution.
>
> **W3**: According to Table 3, compared with Qwen-7B, larger and more capable models seem to approach the performance achieved by introducing ILP (Custom-CoT). Therefore, with the inclusion of stronger models (such as GPT-5), the effectiveness and significance of this work appear to diminish. However, the paper lacks corresponding experiments to substantiate this observation.
>   - Our experimental focus is not to compete with arbitrarily strong base models, but to study how much formal induction and verification can improve a given MLLM across model scales. On ARC-AGI-1, Table 3 evaluates three strong MLLMs—GPT-4o, Gemini-2.0 Flash, and Qwen-Max—and, for each of them, ILP-CoT improves both accuracy and Hamming distance over Direct Predict and Custom CoT. On 1D-ARC with pure textual Qwen3-8B and Qwen3-14B, ILP-CoT again consistently improves performance, with larger relative gains for the smaller model but still clear gains for the larger one.  These results indicate that ILP-CoT provides complementary benefits—especially in structural consistency—even as base models become stronger. Moreover, the value of ILP-CoT is not only in raw task accuracy, but also in interpretable, verifiable reasoning. ILP-CoT always outputs executable Prolog-style rules, treats negative examples as hard logical constraints, and uses ILP verification to prune hallucinated hypotheses and repair missing conditions.
>   - Conceptually, extremely large models like GPT-5 can indeed solve some instances without explicit ILP, but they do not by themselves guarantee symbol-level faithfulness or provide an executable rule set. Our contribution is to offers a training-free way to turn a wide range of MLLMs—small or large—into systems that perform abductive rule induction with explicit logical guarantees.

---

> > ### Comment · Reviewer_Hxhn · 2025-11-26
> > **Official Comment by Reviewer Hxhn**
> >
> > Thank you for the response, which largely resolved my previous confusion. However, I still have the following questions:
> >
> > 1. The main experimental results of the work are still unconvincing to me. I understand the lack of cutting-edge methods specifically designed for multimodal inductive reasoning, but I believe the authors should explore the transfer of other CoT-based methods to the multimodal domain, such as SC [1] and Self-Refine [2]. Furthermore, there are also representative methods in the field of pure language inductive reasoning [3,4], which I believe would not be difficult to transfer to the dataset of this work. I believe the inclusion of these works is necessary, as it would demonstrate that the problem claimed in this paper cannot be adequately solved by these previous classic methods.
> > 2. Additionally, based on my understanding, the proposed method is plug-and-play. Therefore, it should be deployed on a wider variety of base models to prove that this performance improvement is independent of the base model, rather than merely experimenting only on Qwen under CLEVR-Hans.
> > 3. To my knowledge, there are already many works in the field of logical reasoning that combine LLMs with LP languages [5,6]. Given that your work falls under logical reasoning and your method also involves a combination with LP, please explain the innovation of your work compared to these related works.
> >
> >
> >
> > [1] Wang X, Wei J, Schuurmans D, et al. Self-consistency improves chain of thought reasoning in language models[J]. arXiv preprint arXiv:2203.11171, 2022.
> >
> > [2] Madaan A, Tandon N, Gupta P, et al. Self-refine: Iterative refinement with self-feedback[J]. Advances in Neural Information Processing Systems, 2023, 36: 46534-46594.
> >
> > [3] Qiu L, Jiang L, Lu X, et al. Phenomenal yet puzzling: Testing inductive reasoning capabilities of language models with hypothesis refinement[J]. arXiv preprint arXiv:2310.08559, 2023.
> >
> > [4] Wang R, Zelikman E, Poesia G, et al. Hypothesis search: Inductive reasoning with language models[J]. arXiv preprint arXiv:2309.05660, 2023.
> >
> > [5] Pan L, Albalak A, Wang X, et al. Logic-lm: Empowering large language models with symbolic solvers for faithful logical reasoning[C]//Findings of the Association for Computational Linguistics: EMNLP 2023. 2023: 3806-3824.
> >
> > [6] Olausson T, Gu A, Lipkin B, et al. LINC: A neurosymbolic approach for logical reasoning by combining language models with first-order logic provers[C]//Proceedings of the 2023 Conference on Empirical Methods in Natural Language Processing. 2023: 5153-5176.

---

> ### Author Response · Authors · 2025-12-01
>
> **We would like to thank the reviewer for the further comments. We have fully covered the discussions on the suggested works in the revision.**
>
> - **Q1**: Considering suggested works [1-4].
>   - We thank the reviewer for suggesting these approaches. For [1] [2], we kindly note that they are designed for deductive reasoning tasks such as math QA. It is invalid to directly apply them to inductive reasoning tasks studied in our work. Furthermore, these methods rely on MLLM themselves to detect their errors and reflect. As shown by our failure reflection analysis in Sec. 4.6, MLLM is hard to realize their failures by themselves in induction tasks, possibly due to strong perceptual hallucinations. While ILP can serve as an effective and efficient tool for failure detection and reflection.
>   - For [3][4], we have conducted further experimental comparison in Sec. B. We report the performance comparison over [3][4] under the ARC-AGI benchmark. We utilize GPT-4o as the base model and test under the full 400 examples in its training set. The results show the desirable performance of ILP-CoT.
>
> - **Q2**: Additionally, based on my understanding, the proposed method is plug-and-play. Therefore, it should be deployed on a wider variety of base models to prove that this performance improvement is independent of the base model, rather than merely experimenting only on Qwen under CLEVR-Hans.
>   - The experiments are carefully designed  to introduce various benchmarks (CLEVR-Hans, ARC-1D, ARC-AGI, Text-to-Image Customization), ranging from textual to visual inductive reasoning with different difficulties. We have also tested various small/large close/open-source MLLMs to justify the generality of our method. The choice is based on their representativeness, as well as the fitness to the difficulties of the benchmarks.
>
> - **Q3**: Considering suggested works [5][6].
>   - We kindly note that the suggested works [5][6] study deductive logic reasoning, which is a significantly different task to abductive logical rule induction studied in our work. We have discussed in Sec. 2: deductive reasoning requires the ability of conducting long-chain reasoning, while focus less on correctly perceiving and understanding the complicated inputs, which is the focus of inductive reasoning. Directly applying the techniques in deductive reasoning is invalid for induction. On the other hand, our work proposes a systematic solution to integrate ILP with MLLMs, tacking the central technical challenges in achieving this. We believe that our method can inspire future research of building powerful inductive reasoning methods bridging neural and symbolic systems.

---

### Official Review · Reviewer_apT9 · 2025-10-31

**Soundness:** 2
**Presentation:** 2
**Contribution:** 3
**Rating:** 4
**Confidence:** 3

**Summary:**

This paper proposes ILP-CoT, a framework that bridges MLLMs with ILP to induce interpretable and verifiable logical rules from unstructured textual or visual inputs. The MLLM processes raw inputs to generate symbolic facts and structure-consistent candidate rules, which are then translated into meta-rules for ILP. A reflection loop detects inconsistencies and triggers regeneration on the MLLM side. The authors provide multi-domain evaluations and release open-source code to support reproducibility.

**Strengths:**

1. This paper introduces a useful, verifiable pipeline that connects raw inputs to symbolic reasoning via LLM and ILP; the plug-in design supports different MLLMs without task-specific training, offering good flexibility.

2. The approach uses the observation that MLLMs can often produce structure-correct rule forms even with hallucinations, although this assumption depends on the quality of the base model, the authors also provides a failure reflection loop to ensure the automatically ILP tasks can be automatically constructed without further training.

3. The evaluation section presents multi-domain results, showing that ILP-CoT outperforms Direct-Predict and Custom-CoT under the same base model.

**Weaknesses:**

1. Time and computational cost: the cost of ILP-CoT is not clearly quantified, it would be helpful to report the average time for MLLM generation, reflection-loop retries (which depend on base-model quality and task complexity but could be summarized as average time), and ILP solving. The authors note that failure reflections can be expensive for large numbers of subjects, but the costs and cost–accuracy trade-offs for smaller or typical settings are also unclear.

2. Fairness under reflection: it is unclear whether sampling budgets are aligned across Direct-Predict, Custom-CoT, and ILP-CoT, since ILP-CoT includes failure reflection retries, the evaluation should clarify whether the other methods use the same number of decoding attempts, ensuring that the reported gains are not simply due to additional MLLM attempts.

3. Base-model performance discussion: different base models are used across tasks (e.g., Qwen-7B for CLEVR-Hans; GPT-4o, Gemini-2.0 Flash, and Qwen-Max for ARC), but the reason for these selections is not stated. A brief discussion of this selection would help, since the performance of this method clearly depends on base-model quality: for example, in table 2 the improvement over Custom-CoT is small (~0.25 %) in Qwen-max compared with other methods. In table 1 the fully trained Neumann model outperforms ILP-CoT, while this may come from pretraining of Neumann on this task, clarifying whether a stronger base MLLM could close this gap would better illustrate the trade-off between training a CNN and using a pretrained MLLM without fine-tuning to achieve comparable results.

**Questions:**

1. Please clarify or align the sample/retry budget across Direct-Predict, Custom-CoT, and ILP-CoT (e.g., ensure all results are reported under similar MLLM settings, exact@1 or pass@k)?

2. Could you quantify the relation between #reflections and accuracy, or the overall time and accuracy?

3. How different MLLMs affect the performance in each tasks, and to what extent MLLMs without pretraining can outperform pretrained CNNs in terms of cost or accuracy?

---

> ### Author Response · Authors · 2025-11-25
> **Author Response**
>
> **We would like to thank the reviewer for the detailed comments**.
>
> **W1**: Time and computational cost: the cost of ILP-CoT is not clearly quantified.
>   - We have included Sec. 4.6 to discuss time cost. We set a maximum limit of 5 reflection loops to prevent indefinite execution, and on average, the system requires more than 2 reflection iterations to successfully induce a valid rule. In terms of specific runtime for each stage, the fact-capturing process performed by the MLLM typically takes between 5 to 7 seconds. For the logical rule induction, we enforce a strict timeout of 20 seconds for the ILP solver, though in practice the solving process usually completes within 8 to 12 seconds. From these statistics, we can see that ILP-CoT introduces minor computational overhead w.r.t. its significant performance gains.
>
> **W2**: Fairness under reflection.
>   - We have included Sec. 4.6 to discuss the reflection mechanism. The key is that we do not explicitly set the number of retries for each method. The reflection is triggered only when the method can be aware of its failure. In contrast to other baselines, in which the reflection should be decided by the MLLM itself, ILP-CoT incorporates reflection triggered by the formal detection of failures during rule induction.
>
>   - To justify the advantage of utilizing ILP for failure reflection, we report reflection triggering rates of GPT-4o under the ILP-CoT-Customization 3P3N experiment in Tab.8. The reflection loop itself is a key contributor to ILP-CoT's robustness. 87.2% of cases enter the loop exactly because no consistent rule is learned initially. When we replace the ILP guided trigger with purely LLM-based self-evaluation, GPT-4o never flags its own rules as incorrect when judging only on positive examples, and when given both positive and negative examples it chooses to restart on only 6.3% of cases. In contrast, the ILP-guided trigger reliably detects inconsistent rules and activates the reflection loop on precisely those instances where the current hypothesis is provably inadequate. Combined with the verify–revise procedures, this targeted triggering turns reflection into a principled debugging mechanism, yielding substantially higher coverage and precision with minimal additional cost.
>
> **W3**: Base-model performance discussion.
>   - Our goal is to study how much ILP-CoT can improve over a given base MLLM, rather than to optimize for task-specific SOTA. For CLEVR-Hans, we use the open-weight Qwen-7B vision–language model so that the full ILP-CoT pipeline can be run without task-specific finetuning and in a reproducible way. For ARC-AGI, we deliberately choose three diverse, strong MLLMs (GPT-4o, Gemini-2.0 Flash, Qwen-Max) to test whether ILP-CoT yields consistent gains across different model families, and for 1D-ARC we include Qwen3-8B/14B to show that the same pattern holds for smaller, purely textual models. Across these settings ILP-CoT consistently improves both accuracy and Hamming distance over Direct-Predict and Custom-CoT.
>
>   - On CLEVR-Hans, the fully trained Neumann model still attains the highest accuracy, but it relies on a task-specific CNN perceptual module pre-trained on the CLEVR dataset. In contrast, ILP-CoT is training-free: it uses an off-the-shelf MLLM without any finetuning and constructs ILP problems on the fly. Under these much weaker assumptions, ILP-CoT substantially outperforms NEUMANN without pretraining and approaches the fully trained NEUMANN model.
>
> **Q1 & Q2**: Please clarify or align the sample/retry budget across Direct-Predict, Custom-CoT, and ILP-CoT (e.g., ensure all results are reported under similar MLLM settings, exact@1 or pass@k)? Could you quantify the relation between #reflections and accuracy, or the overall time and accuracy?
>   - Please see the responses to **W1** and **W2**.
>
> **Q3**: How different MLLMs affect the performance in each tasks, and to what extent MLLMs without pretraining can outperform pretrained CNNs in terms of cost or accuracy?
>   - Please see the response to **W3**.

---

### Official Review · Reviewer_mNWp · 2025-11-09

**Soundness:** 3
**Presentation:** 3
**Contribution:** 3
**Rating:** 4
**Confidence:** 3

**Summary:**

This work proposes a neuro-symbolic framework that combines multimodal large language models (MLLMs) and inductive logic programming (ILP).
The MLLMs provide powerful perception ability, including identify entities and relations from images, corresponding to logical facts.
ILP can produces interpretable and verifiable logical rules according to the logical facts.
Experiments show the effectiveness of the proposed method.
Authors also propose text-to-image customized generation, a potential application of the framework.

**Strengths:**

1. The paper is well written and formatted.
2. Authors propose a partial interpretable neuro-symbolic architecture, which is a kind of frontier research.
3. Experiments are well designed and can justify the main claims.

**Weaknesses:**

1. The definition of ILP in Section 3.1 is not clear enough. For example
    - What exactly is $H, B, E^+, E^-$? It seems that they are set of Horn clauses, which is a common but not unique setting in ILP.
    - Are $E^+$ and $E^-$ sets of logical facts? If so, the formula $H \cup B \models E^+, H \cup B \nvDash E^-$ is not clear since the entailment consequence can not be a set.
    - Inductive bias can mean many things in ILP, e.g., language bias. What exactly $B$ specifies?

    I suggest rewriting the definition of ILP in a more mathmatical and rigorous manner. Appropriate citations or examples may help. Otherwise, it is hard to understand the exact problem scope authors are tackling.
2. Section 3.3: "Generating rule structure proposals using MLLM" is not well motivated. "MLLMs could propose structure-correct rules even under hallucinations" is just an observation but not a motivation, since proposing structure-correct rules does not imply proposing helpful meta-rules.
3. Section 4.5: The ablation study is not fair. The proposed method defines inductive bias through meta-rules for Metagol, but does not provide inductive bias for Popper and ILASP. Although Popper and ILASP employ other format of inductive bias, inductive bias is crucial for symbolic ILP approaches. To conduct a fair comparison, I suggest providing inductive bias for Popper and ILASP in some other ways.
4. Combining neural perception and ILP is not a new idea but authors do not mention some existing methods. Authors should discuss them or conduct experiments on them. A non-exhaustive references is given below.


[1] Robin Manhaeve, Sebastijan Dumancic, Angelika Kimmig, Thomas Demeester, and Luc De Raedt. Deepproblog: Neural probabilistic logic programming. In S. Bengio, H. Wallach, H. Larochelle, K. Grauman, N. Cesa-Bianchi, and R. Garnett (eds.), Advances in Neural Information Processing Systems, volume 31. Curran Associates, Inc.,
2018.

[2] Wang-Zhou Dai and Stephen Muggleton. Abductive knowledge induction from raw data. In Zhi-Hua Zhou (ed.), Proceedings of the Thirtieth International Joint Conference on Artificial Intelligence, IJCAI-21, pp. 1845–1851. International Joint Conferences on Artificial Intelligence Organization, 8 2021.

[3] Shindo, H., Pfanschilling, V., Dhami, D.S. et al. αILP: thinking visual scenes as differentiable logic programs. Mach Learn 112, 1465–1497 (2023). https://doi.org/10.1007/s10994-023-06320-1

**Questions:**

1. Please explain Section 3.2 by throughout examples or formal statement. Especially, what is activation schema? Is it designed manually? What is single-state & multi-state?

---

> ### Author Response · Authors · 2025-11-25
> **Author Response**
>
> **We would like to thank the reviewer for the detailed comments**.
>
> **W1**: The definition of ILP in Section 3.1 is not clear enough.
>   - Our definition follows the standard ILP literature such as [Cropper & Dumancic, 2022] cited in the paper. We have revised Section 3.1 to further make the definitions more concrete.
>
> **W2**: "Generating rule structure proposals using MLLM" is not well motivated.
>   - We kindly note that both "MLLMs can propose structure-correct rules under hallucinations" and " this leads to helpful meta-rules" are solid arguments. For the first argument, we include an additional experimental analysis on the proportion that MLLMs hallucinate on facts but still generate structure-correct rules, as in Tab. 8 and discussed in Sec. 4.4. This soundly verifies our observations. For the second argument, we have detailedly discussed in Sec. 3.3 that "The obtained meta-rules then serves as a strong structural bias for Metagol, directly constraining admissible rule forms and the corresponding search space, thereby producing efficient, interpretable candidates and enabling rapid convergence in the ILP step".
>
> **W3**: Section 4.5: The ablation study is not fair.
>   - We kindly note that the inductive biases generated by MLLM are also provided to POPPER and ILASP in our experiments. We further clarify this in Sec. 4.5. The reason that they perform worse is not that less information from MLLM is provided to them, but their native representation of such information is at a lower-level than meta-rules, making it less effective in pruning the search space. This is discussed in detail in Sec. 3.3.
>
> **W4**: Combining neural perception and ILP is not a new idea but authors do not mention some existing methods.
>   - We kindly note that the first two suggested works are actually cited in the paper (See Sec. 1). The third work (alphaILP) is actually the prior work of NEUMANN, which is cited and compared in our work. We have further cited alphaILP in the revision.
>
> **Q1**: Section 3.2 is not sufficiently clear.
>   - We have revised Sec. 3.2 for better clarity.

---

> > ### Comment · Reviewer_mNWp · 2025-11-27
> >
> > > Our definition follows the standard ILP literature such as [Cropper & Dumancic, 2022] cited in the paper. We have revised Section 3.1 to further make the definitions more concrete.
> >
> > - [Cropper & Dumancic, 2022] presents two learning settings. Which one do you use?
> > - In the revised manuscript, you said "Given these sets, an ILP system induces H ∈ H that is consistent with B". However, the consistency defined in [Cropper & Dumancic, 2022] states about the entailment relation between H,B and $E^-$. These are two different aspects of consistency, and I've never seen the definition you use.
> >
> >
> >
> > > We kindly note that both "MLLMs can propose structure-correct rules under hallucinations" and " this leads to helpful meta-rules" are solid arguments. For the first argument, we include an additional experimental analysis on the proportion that MLLMs hallucinate on facts but still generate structure-correct rules, as in Tab. 8 and discussed in Sec. 4.4. This soundly verifies our observations. For the second argument, we have detailedly discussed in Sec. 3.3 that "The obtained meta-rules then serves as a strong structural bias for Metagol, directly constraining admissible rule forms and the corresponding search space, thereby producing efficient, interpretable candidates and enabling rapid convergence in the ILP step".
> >
> > - The obtained meta-rules do serve as a strong structural bias. However, no prior knowledge justifies that this bias is correct.
> >
> >
> >
> >
> > Thanks to authors' further explanation, I understand more things, but one more concern arises.
> > 1. According to Table 1, ILP-CoT performs worse than NEUMANN (with pretraining). The MLLM used in ILP-CoT has also been pretrained for perception. Hence, authors should compare ILP-CoT with NEUMANN (with pretraining).

---

> > > ### Author Response · Authors · 2025-12-01
> > >
> > > **We would like to thank the reviewer for the further comments.**
> > >
> > > - **Q1**: Further question for Sec. 3.1.
> > >   - We have further revised Sec. 3.1 for better clarity.
> > >
> > > - **Q2**: The obtained meta-rules do serve as a strong structural bias. However, no prior knowledge justifies that this bias is correct.
> > >   - We kindly note that it would be unrealistic to ask for absolute correctness of the generated meta-rules. This is true even when the meta-rules are designed by human experts. Our major contribution lies in that we have designed an automated way of generating meta-rules with high chances to be correct. As discussed in [Cropper & Dumancic, 2022], this is an important problem in ILP which is "largely under-researched".
> > >
> > > - **Q3**:  According to Table 1, ILP-CoT performs worse than NEUMANN (with pretraining). The MLLM used in ILP-CoT has also been pretrained for perception. Hence, authors should compare ILP-CoT with NEUMANN (with pretraining).
> > >   - We kindly note that in NEUMANN with pre-training, its perception module is a slot-attention model that is pre-trained under the CLEVR dataset, but not pre-trained on any other datasets. The slot-attention model has specific structure to extract symbolic concepts in CLEVR images. This is obviously not the case for MLLM, whose visual encoder is not specifically designed and trained for perceiving CLEVR images. As a result, the chance of hallucination for MLLM is significantly higher than NEUMANN's pre-trained visual perception module. We have shown that ILP-CoT enables the MLLM to achieve desirable performance even under this significantly disadvantaged situation.

---

### Note · Authors · 2026-01-20

I have read and agree with the venue's withdrawal policy on behalf of myself and my co-authors.